# DHODH Inhibition Exerts Synergistic Therapeutic Effect with Cisplatin to Induce Ferroptosis in Cervical Cancer through Regulating mTOR Pathway

**DOI:** 10.3390/cancers15020546

**Published:** 2023-01-16

**Authors:** Mengying Jiang, Yizuo Song, Hejing Liu, Yanshan Jin, Ruyi Li, Xueqiong Zhu

**Affiliations:** Center for Uterine Cancer Diagnosis & Therapy Research of Zhejiang Province, Department of Obstetrics and Gynecology, The Second Affiliated Hospital of Wenzhou Medical University, Wenzhou 325027, China

**Keywords:** DHODH, cervical cancer, ferroptosis, brequinar, cisplatin, mTOR

## Abstract

**Simple Summary:**

Ferroptosis exhibits potent antitumor ability and dihydroorotate dehydrogenase (DHODH) has recently been identified as a novel ferroptosis defender independently of GPX4 or FSP1. DHODH has been studied in several diseases including oral squamous cell carcinoma and small cell lung cancer. This study demonstrates that DHODH inhibition inhibits cervical cancer cells’ cell proliferation and induces cell death through ferroptosis. Moreover, the combination of DHODH inhibition and cisplatin synergistically inhibits the growth of cervical cancer cells in vitro and in vivo by ferroptosis via the mTOR pathway.

**Abstract:**

Ferroptosis exhibits a potent antitumor effect and dihydroorotate dehydrogenase (DHODH) has recently been identified as a novel ferroptosis defender. However, the role of DHODH inhibition in cervical cancer cells is unclear, particularly in synergy with cisplatin via ferroptosis. Herein, shRNA and brequinar were used to knock down *DHODH* and directly inhibit DHODH, respectively. Immunohistochemistry and Western blotting assays were performed to measure the expression of proteins. CCK-8 and colony formation assays were employed to assess the cell viability and proliferation. Ferroptosis was monitored through flow cytometry, the malondialdehyde assay kit and JC-1 staining analyses. The nude mouse xenograft model was generated to examine the effect of combination of DHODH inhibition and cisplatin on tumor growth in vivo. The expression of DHODH was increased in cervical cancer tissues. DHODH inhibition inhibited the proliferation and promoted the ferroptosis in cervical cancer cells. A combination of DHODH inhibition and cisplatin synergistically induced both in vitro and in vivo ferroptosis and downregulated the ferroptosis defender mTOR pathway. Therefore, the combination of DHODH inhibition and cisplatin exhibits synergistic effects on ferroptosis induction via inhibiting the mTOR pathway could provide a promising way for cervical cancer therapy.

## 1. Introduction

The WHO has proposed an action to eliminate cervical cancer by 2030 as a global public health issue [1]. However, it is difficult to achieve the goal for developing countries and regions in the short term. This is mainly attributed to the regional inequalities in the human papillomavirus (HPV) vaccination and screening uptake in developing countries and regions [2], which have made cervical cancer continue to be the most common and deadly gynecologic malignancies during the past decade [3]. Moreover, the morbidity and mortality of cervical cancer show a younger annual trend [4]. Hence, cisplatin-based chemotherapy has been increasingly used for managing early-stage cervical cancer owing to the raising demand for fertility preservation [5]. However, notorious multi-organ toxicities and drug resistance often lead to the poor prognosis [6,7]. Therefore, novel therapeutic strategies with more efficacy and safety are urgently needed for cervical cancer therapy.

Ferroptosis, a novel form of regulated cell death, is distinct from the traditional defined cellular apoptosis and necrosis. This iron-dependent cell death is initiated and cascaded by excessive lipid peroxidation and reactive oxygen species (ROS) production [8]. In particular, several tumor suppressors (e.g., p53, BAP1) could inhibit cystine uptake by repressing the expression of cystine transporter SLC7A11, promoting ferroptosis-induced cancer cell death upon lipid peroxidation accumulation. This evidence has validated the tumor suppressive role of ferroptosis in cancer development [9,10]. In addition, elevated iron dependency is reported to retain normal growth more in cancer cells than normal cells, which endows cancer cells with increased vulnerability to ferroptosis [11]. Thus, targeting ferroptosis activation is a promising way to kill cancers more precisely. There are two classic antioxidant defense systems to attenuate ferroptosis, including GPX4 and FSP1. Specifically, GPX4 reduces the glutathione (GSH) level to detoxify the phospholipid hydroperoxides and inhibits ferroptosis [12]. FSP1 serves as an oxidoreductase in parallel to GPX4 to detoxify lipid peroxyl radicals and restrain ferroptosis [13,14]. Conversely, ferroptosis can be induced by disrupted GPX4 or FSP1 defense function, thereby playing the anticancer effects via excessive ROS formation [15,16,17].

Dihydroorotate dehydrogenase (DHODH) is a key enzyme catalyzing the de novo pyrimidine synthesis and is located in the inner membrane of the mitochondria [18]. Strikingly, DHODH has been recently identified as a novel defender to inhibit ferroptosis by reducing ubiquinone to ubiquinol, which is independent of GPX4 or FSP1 [19]. Indeed, the determined tumorigenic role of DHODH has led to numerous attempts to develop DHODH inhibitors for cancer treatment [20]. Brequinar (BQR), a common and commercial DHODH inhibitor, has been tested to exert a potential anticancer effect in multiple preclinical studies for decades [21,22]. However, little is known about whether brequinar could kill cancer cells by increasing DHODH-mediated ferroptosis. Moreover, whether inhibition of DHODH could exert a synergistic function with cisplatin for treating human cancers remains undefined.

Herein, using an immunohistochemistry (IHC) staining assay, we detected the DHODH expression in tissues of cervical cancer. Then, the *DHODH*-silenced cell lines were generated and analyzed for ferroptosis in cervical cancer cells. Moreover, both in vitro and in vivo studies were performed to investigate the synergistic anticancer effects of DHODH inhibition and cisplatin by inducing ferroptosis. Finally, the underlying mechanism through which DHODH inhibition and cisplatin exerted synergistic function was studied. This study proposed that the combination of DHODH inhibition and cisplatin offers a potential treatment for cervical cancer.

## 2. Materials and Methods

### 2.1. Reagent and Antibodies

Brequinar and liproxstatin-1 (Lip-1) were obtained from MedChemExpress (Monmouth Junction, NJ, USA). Cisplatin (DDP) was acquired from Sigma (St. Louis, MO, USA). Brequinar was prepared in a range of concentrations (0.01, 0.1, 1, 10 and 100 μM) using dimethylsulfoxide (DMSO) and stored at −20 °C before use. CCK-8 kit was purchased from Dojindo (Kumamoto, Japan). The BCA protein assay kit was purchased from Beyotime (Shanghai, China). Primary antibodies for rabbit anti-human GAPDH, rabbit anti-human mTOR and rabbit anti-human p-mTOR were purchased from CST (Danvers, MA, USA). Primary antibodies for rabbit anti-human vinculin, rabbit anti-human Ki-67 and mouse anti-human 4-HNE were purchased from Abcam (Cambridge, UK). Primary antibody for rabbit anti-human DHODH was purchased by ABclonal (Wuhan, China). The second goat anti-rabbit and anti-mouse antibodies conjugated with peroxidase were purchased from Biosharp (Shanghai, China). DAB Substrate Kit was obtained from Beijing Zhongshan Golden Bridge Biotechnology Co., Ltd. (Beijing, China). The hematoxylin staining solution was obtained from Solarbio (Beijing, China). DAPI solution was purchased from Solarbio (China). The TUNEL detection kit was acquired from Roche (Basel, Switzerland).

### 2.2. Immunohistochemistry

The procedure related to human subjects was approved by the Ethics Committee of The Second Affiliated Hospital of Wenzhou Medical University (2022-K-141-02). The cervical cancer and matched normal tissues embedded in paraffin were obtained from the Second Affiliated Hospital of Wenzhou Medical University. The tissues were cut into 5 μm thickness pieces. After deparaffinization in xylene, the pieces were rehydrated in ethanol gradients. Then the sections were microwaved for antigen retrieval and immersed in 3% hydrogen peroxide. After being blocked by 5% BSA, incubation with DHODH (1:200) antibodies was performed overnight at 4 °C. A secondary antibody was incubated on the slides the following day after washing with phosphate buffer saline (PBS). Then, the slides were stained with DAB solution. The nuclear detection was enhanced by using hematoxylin solution. Finally, the slides were observed under the microscope (Leica, Wetzlar, Germany). The immune scores of each slide were evaluated and calculated as previously described [23].

### 2.3. Cell Culture

Human cervical adenocarcinoma HeLa cells and human cervical squamous cell carcinoma CaSki cells were provided from ATCC (Manassas, VA, USA). HeLa cells were grown in Dulbecco’s Modified Eagle Medium (DMEM; Gibco, Grand Island, NY, USA) with 10% fetal bovine serum (FBS; Gibco, Grand Island, NY, USA) and 1% penicillin–streptomycin (Pen–Strep) solution (Invitrogen, Carlsbad, CA, USA). A Roswell Park Memorial Institute 1640 (RPMI-1640) medium with 10% FBS and 1% Pen-Strep solution was used to maintain CaSki cells. All cell lines were incubated at 37 °C in a humidified atmosphere with 5% CO_2_.

### 2.4. Cell Transfection

For the *DHODH* knockdown in HeLa and CaSki cells, the shRNA sequences were designed and then synthesized by Sigma-Aldrich (Shanghai, China) and were cloned into the pLKO.1-TRC vector (Origene, Rockville, MD, USA). The pLKO.1-TRC vector was employed as an empty control vector. After confirming the gene sequence, packing vectors psPAX2 and pMD2.G were used to pack the virus in 293T cells. Finally, the harvested lentiviral particles were filtered to transfect CaSki and HeLa cells.

### 2.5. Cell Viability Assay

Cells were plated onto 96-well plates and cultured overnight with 2 × 10^3^ cells per well. Cells were incubated for a set of predefined times (0, 24, 48 and 72 h). At 37 °C, cells were incubated for another 2 h after CCK-8 reagent (10 μL/well) was added. Cell viability was determined at 450 nm using a microplate reader (Thermo Fisher Scientific, Waltham, MA, USA).

### 2.6. Drug Sensitivity Analysis

In 96-well plates, cells were seeded and cultured for 24 h. Then, various concentrations of cisplatin (0, 1, 2, 4, 8 and 16 μM) and brequinar (0.01, 0.1, 1, 10 and 100 μM) were added. Subsequently, a second incubation at 37 °C was performed after exposure to 10 μL CCK-8 reagent. The absorbance was measured at 450 nm in a microplate reader. For calculating the inhibition rate, the following formula was used: Inhibition rate (%) = [(OD_450_ of control well − OD_450_ of test well) ÷ (OD_450_ of control well − OD_450_ of blank well)] × 100%. The combination index (CI) value was calculated using CompuSyn software to investigate the drug-drug interaction between brequinar and cisplatin. The two drugs were synergistic in killing cells when the CI value was between 0 and 1. It indicated a stronger synergistic effect of the two drugs if the value was closer to one.

### 2.7. Colony Formation Assay

In brief, HeLa and CaSki cells (1 × 10^3^ cells/well) were plated into 6-well plates and cultured for a week. Fixed cells were stained with 0.25% crystal violet (Beyotime, China) for 10 min after fixation in 4% paraformaldehyde. Finally, the colonies were observed under a microscope.

### 2.8. Flow Cytometric Analysis

Propidium iodide (PI) staining (BD Biosciences, Franklin Lake, NJ, USA) was used to determine cell death by flow cytometry. CaSki and HeLa cells were plated at the 3 × 10^5^ cells/well density into 6-well plates and cultured for 24 h. Following the treatment with different reagents, the cells were harvested, washed in cold PBS and stained with 5 μg/mL PI for 15 min at dark. The percentage of the PI-positive dead cell population was analyzed using CytoFLEX flow cytometry (Beckman, Bria, CA, USA).

### 2.9. Western Blotting Analysis

Cells were collected after ice-cold lysis with a radio-immunoprecipitation assay buffer containing 1 mM phenylmethanesulfonyl fluoride (PMSF; Beyotime, China) for 30 min. A centrifugation of 12,000× *g* at 4 °C for 20 min was performed on lysates. The supernatant was collected for protein quantification by a BCA assay. SDS-PAGE was used for2 electrophoresis of equal amounts of proteins, followed by polyvinylidene fluoride (PVDF) membrane transfer (Millipore, Boston, MA, USA). Incubation with the primary antibodies was performed overnight at 4 °C after blocking with 5% non-fat milk. Afterwards, the membranes were incubated at room temperature for 1 h with the secondary antibody. The concentration of antibodies used in this study was listed as follows: DHODH (1:1000), p-mTOR (1:1000), mTOR (1:1000), GAPDH (1:5000), and vinculin (1:2000). Using ECL reagent (Beyotime, China), protein bands were visualized. The densitometry readings of each band were calculated by ImagJ software 1.8.0. (NIH, Bethesda, MD, USA). The whole blots with all the bands with all molecular weight markers are provided in Appendix A.

### 2.10. Lipid Peroxidation Measurement

Lipid peroxidation was measured using the malondialdehyde (MDA) assay kit (Solarbio, China). In brief, cells were seeded on 6-well plates at 5 × 10^5^ cells per well after which they were treated with either brequinar (1 μM) or cisplatin (2 μM) for 48 h. Then, cells were harvested and resuspended by MDA extracting solution. By centrifugation at 8000× *g* for 10 min under 4 °C, the supernatant was collected. The protein concentration of the lysis sample was tested by BCA and the rest of the lysis sample was mixed with working solution and incubated at 100 °C for 1 h. All mixtures were centrifugated at 10,000× *g* for 10 min. Measurements were made at 532, 450 and 600 nm for absorbance of the supernatant. The MDA content was calculated based on manufacturer’s instructions.

### 2.11. JC-1 Mitoscreen Assay

A total of 2 × 10^5^ CaSki or HeLa cells were seeded on 6-well plates overnight after which they were treated with either brequinar (1 μM) or cisplatin (2 μM). By using the mitochondrial membrane potential (MMP) kit with JC-1 (Solarbio, Beijing, China), MMP level was detected after the treatment with reagents for 48 h. Images of stained cells were obtained using a fluorescence microscope or analyzed them with flow cytometry.

### 2.12. Animal Studies

Five-week-old female BALB/c nude mice (Beijing Weitonglihua Sciences Co., Ltd., Beijing, China) were housed under a specific pathogen-free (SPF) condition with sterilized food and water. After nude mice adapted for ~1 week, 2 × 10^6^ HeLa cells were subcutaneously injected with 100 μL PBS in the right flank of mice. Upon reaching 50 mm^3^ of tumor size, mice were randomly assigned into four groups (*n* = 8 per group). The treatments in each group were as follows: (i) control (sterilized saline); (ii) brequinar group (15 mg/kg BQR); (iii) cisplatin group (7 mg/kg DDP); (iv) brequinar + cisplatin group (15 mg/kg BQR + 7 mg/kg DDP). Brequinar was dissolved in DMSO and diluted in saline containing PEG400, then intraperitoneally injected every three days. DDP was formulated in saline and administered intraperitoneally once a week. A body weight measurement was obtained every three days, as well as a tumor volume. A formula was used to calculate tumor volume: volume = length × width^2^ × 1/2. The mice were sacrificed at day 12. The tumor tissues were collected for making tissue sections. The in vivo antitumor and ferroptosis induction abilities of different treatments were evaluated by IHC (Ki-67 (1:200), 4-HNE (1:200)) and TUNEL staining assays as directed by the manufacturer. Animal experiments were approved by the Institutional Animal Care and Use Committee of Wenzhou Medical University (wydw2022-0561).

### 2.13. Statistical Analysis

Prior to comparison, Kolmogorov–Smirnov analysis was performed on all data. Normally distributed data were presented as means ± SD. Differences were compared using Student’s *t*-test (two groups) and ANOVA (more than three groups). The least significance method was used if variances were homogeneous. Dunnett’s T3 method was employed if variances were nonhomogeneous. Statistical significance was determined by a *p*-value < 0.05.

## 3. Results

### 3.1. DHODH Inhibition Suppresses the Proliferation and Promotes the Death in Cervical Cancer Cells

Initially, we compared the expression of DHODH protein expression in tissues of cervical cancer with adjacent normal tissues using an IHC assay. As illustrated in Figure 1A, a significant increase in DHODH expression was observed in cervical cancer tissues but not in normal tissues. Next, for further verification of the carcinogenic effect of DHODH, two sets of shRNA sequences (sh1 and sh2) were employed to knock down *DHODH* in HeLa and CaSki cells. Results from Western blotting verified the success of *DHODH* silence in these two cell lines (Figure 1B and Appendix A).

Further experiments were carried out on CCK-8 and colony formation to examine whether DHODH affected the cell viability and proliferation. In comparison with the control group, *DHODH*-silenced CaSki and HeLa cells had significantly reduced cell viability and clonogenicity (Figure 1C,D). In parallel to *DHODH* silence, brequinar, a specific inhibitor of DHODH, was used to suppress the DHODH activity in cervical cancer cells. Brequinar-treated cells were tested for viability and death using CCK-8 and flow cytometry assays. Obviously, brequinar decreased the survival rate in both CaSki and HeLa cells (Figure 1E). As measured in HeLa cells, the IC_50_ values for brequinar were 5.649 mM (24 h), 0.338 μM (48 h) and 0.156 μM (72 h). CaSki cells showed IC_50_ values of 0.747 μM (48 h) and 0.228 μM (72 h) for brequinar. Brequinar also induced more cell death in cervical cancer cells, as evidenced by a 2.94-fold increase in PI positive rates for CaSki cells and a 2.32-fold increase in HeLa cells (Figure 1F). In vitro, either genetic knockdown or pharmacological inhibition of DHODH suppresses the proliferation of cervical cancer cells.

### 3.2. DHODH Inhibition Induces Ferroptosis in Cervical Cancer Cells

Given that DHODH inhibition repressed cervical cancer cells’ proliferation by cell death promotion, the role of DHODH inhibition in inducing ferroptosis was investigated. Since ferroptosis is featured as the outcome of lipid peroxidation accumulation, the level of principal metabolite MDA generated during the process was measured. Genetic silence of *DHODH* significantly increased the MDA level in HeLa and CaSki cells (Figure 2A). Consistently, brequinar-induced inhibition on DHODH activity also led to the elevation of MDA level in both cells (Figure 2B). Moreover, liproxstatin-1, known as a classic ferroptosis inhibitor, was used on brequinar-treated cells. Figure 2C indicated that the brequinar-induced decrease in cell viability was partly rescued by the supplementation of liproxstatin-1 in both CaSki and HeLa cells. Furthermore, a significant decrease in the MDA level was observed after liproxstatin-1 treatment in both cells, compared with the brequinar-treated group (Figure 2D), further validating the occurrence of ferroptosis under DHODH inhibition. The membrane damage caused by lipid peroxidation is characterized as a typical morphologic change in the process of ferroptosis. Considering that DHODH is expressed mainly in mitochondria, the mitochondrial dysfunction was tested by evaluating MMP through JC-1 staining. As presented in Figure 2E, the MMP intensity was dramatically reduced in DHODH-inhibited cells, as manifested by the enhanced green to red fluorescence ratio. Collectively, it appears that DHODH inhibition promotes cervical cancer cell death by inducing ferroptosis.

### 3.3. DHODH Inhibition Synergistically Increases Cisplatin-Mediated Cytotoxicity in Cervical Cancer Cells via Ferroptosis

Based on the ferroptotic role of DHODH inhibition, whether DHODH inhibition could sensitize cervical cancer cells to cisplatin through inducing ferroptosis was further explored. During the 48-h incubation, CaSki and HeLa cells were treated with cisplatin at different concentrations. Figure 3A illustrated that the inhibition rate after cisplatin treatment was remarkably increased on both CaSki and HeLa cells after DHODH downregulation. This implied that the sensitivity to cisplatin was increased by DHODH inhibition in cervical cancer cells.

In parallel, the CI value for cisplatin and brequinar was calculated using Compusyn software. Each CI with different drug concentrations in both CaSki and HeLa cells was less than 1, demonstrating that DHODH inhibition exerted a synergistic anticancer function with cisplatin in cervical cancer cells (Figure 3B). Furthermore, cisplatin treatment increased the level of MDA in CaSki and HeLa cells dose- and time-dependently (Appendix A), presuming that ferroptosis would be involved in the synergetic effect of DHODH inhibition and cisplatin. Thus, cell death was determined by performing PI staining, lipid peroxidation by an MDA assay and mitochondrial dysfunction by JC-1 staining. It was shown that, as compared to the control groups, either cisplatin or brequinar monotherapy was effective for promoting cell death (Figure 3C), mitochondrial dysfunction (Figure 3D) and lipid peroxidation (Figure 3E) in both CaSki and HeLa cells. More importantly, a combination of brequinar and cisplatin contributed to more PI rates, JC-1 intensities and MDA levels than that under monotherapy. Therefore, these results verify that DHODH inhibition can cooperate with cisplatin to inhibiting cervical cancer in a ferroptotic way and the combination exerts a critical therapeutic potential in treating cervical cancer.

### 3.4. DHODH Inhibition Synergizes with Cisplatin to Exhibit Ferroptotic Anti-Cancer Effect In Vivo

To further confirm the synergistic therapeutic effect of DHODH inhibition and cisplatin in vivo, subcutaneous tumor xenograft models were generated using HeLa cells. After 12 days of treatment, the tumor volume in the control group administered with saline reached 740.5 ± 307.4 mm^3^. In contrast, mice receiving the monotherapy with brequinar or cisplatin had smaller tumor volumes than those receiving saline. Moreover, the tumor growth in the mice of the combination group was limited to the greatest extent, as confirmed by the smallest tumor size (Figure 4A,B). Similarly, the mice of the combination group exhibited the lightest weight of tumor mass among four groups (Figure 4C). Apparently, the tumor growth in the combined administration group was inhibited and surprisingly, the tumor in two nude mice of this group even disappeared after treatment. Meanwhile, it was worth noting that no significant weight loss or death occurred in the mice of each group under the administrated dosage of brequinar and cisplatin (Figure 4D). This implied the safety of the administrated dosage of brequinar and cisplatin in this study.

Then, IHC staining analysis was performed on the tumor tissues to evaluate the levels of cell proliferation, apoptosis and lipid peroxidation. As compared to the control group, treatment of brequinar or cisplatin as monotherapy slightly reduced the Ki-67 expression and promoted the cell apoptosis. The combination of these two drugs significantly increased the apoptosis and downregulated the Ki-67 in cancer tissues. Consistently, the combination of brequinar and cisplatin led to a significant upregulation of 4-HNE in cancer cells, indicating more accumulation of lipid peroxidation in this group (Figure 4E). These data in animal studies collectively demonstrate that DHODH inhibition is synergistic with cisplatin to killing cervical cancer cells by inducing ferroptosis in vivo.

### 3.5. Combination of DHODH Inhibition and Cisplatin Induces Downregulation of mTOR Pathway in Cervical Cancer Cells

The underlying mechanism by which ferroptosis is induced after DHODH inhibition and cisplatin treatment was further explored. First, the DHODH expression in CaSki and HeLa cells was detected after treatment with brequinar and/or cisplatin by Western blotting. Notably, the combined administration significantly downregulated DHODH in both cells (Figure 5 and Appendix A). Since mTOR is reported to be a key defender against ferroptosis [24], whether mTOR was involved in the synergistic effect of DHODH inhibition and cisplatin was investigated. As displayed in Figure 5 and Appendix A, although the mTOR level in HeLa cells was slightly elevated with cisplatin treatment, the expressions of p-mTOR and mTOR were remarkably decreased after the combined administration in both CaSki and HeLa cells. In conclusion, the occurrence of significant ferroptosis in cervical cancer cells may be mediated via suppression of mTOR pathway after DHODH inhibition and cisplatin combination treatment.

## 4. Discussion

DHODH is a key enzyme for the de novo biosynthesis of pyrimidine-based nucleotides and serves as a known therapeutic target in various diseases [18]. Accumulated molecular cell biology studies reveal that the inhibition of DHODH depletes intracellular pyrimidine nucleotide pools, thereby leading to cell cycle arrest and sensitization to current chemotherapies in cancer cells [18]. Therefore, drugging *DHODH* has increasingly been proposed as part of combination therapies in cancer treatment. The current study demonstrated that DHODH inhibition suppressed the growth and promoted cell death in cervical cancer via ferroptosis. Furthermore, DHODH inhibition enhanced cervical cancer cell sensitivity to cisplatin through inducing ferroptosis in vitro and in vivo. These results offer promising approaches for the treatment of cervical cancer.

In parallel to the differential expression of DHODH in different cancers, the therapeutic efficacy of DHODH inhibition also varies [25]. For example, lung cancer cells are more sensitive to brequinar than fibrosarcoma cells, for the IC_50_ of brequinar in HT-1080 is 10-fold higher than that of NCI-H226 [19]. Even among different lung cancer cells, the sensitivity to brequinar varies [26]. A high level of DHODH was detected in cervical cancer tissues here. Cervical cancer cells were suppressed in proliferation and induced death after *DHODH* knockdown or brequinar treatment. Ferroptosis is a novel process of regulated cell death and the mechanisms regulating ferroptosis in cells have been extensively explored, such as antioxidant basis, which mainly includes the SLC7A11/GSH/GPX4 axis and FSP1 ubiquinone system [27]. Mao et al. [19] reported that DHODH mediates the ferroptosis defense in mitochondria via the ubiquinone system independently of GPX4 or FSP1 system. In our study, brequinar could directly induce ferroptosis without the assistance of any ferroptosis inducer such as erastin, further implying the independent role of DHODH in regulating ferroptosis. However, several studies have unveiled other mechanisms for controlling ferroptosis in cervical cancer cells. For instance, circEPSTI1 and circACAP2 are both involved in inhibiting ferroptosis via SLC7A11/GPX4 axis [28,29], while circLMO1 and Oleanolic acid promote the ferroptotic process through ASCL4 [30,31]. Therefore, it is proposed that the effect of DHODH inhibition is not confined to GPX4 invalidation, but possesses an apparent synergistic effect with GPX4 inactivation.

Nowadays, non-surgery curative treatment for younger, metastatic or recurrent cervical cancer is also limited. Cisplatin-based chemotherapy is important for these patients, but with serious toxic side effects and frequent drug resistance. Thus, a combination with angiogenesis inhibitor bevacizumab has shown an improved median overall survival in metastatic or recurrent cervical cancer patients [32]. However, the application of these targeted therapies in a clinical setting is still limited. Given that cisplatin could induce ferroptosis through downregulating GPX4 [33], it is presumed that DHODH inhibition can synergize with cisplatin through ferroptosis. The results from in vitro and in vivo studies confirmed the hypothesis. The CI value showed that the synergistic effect in HeLa cells was much more remarkable than that in CaSki cells. Therefore, HeLa cells were chosen for animal experiments. Greater benefits were obtained with combination therapy than monotherapy. Moreover, combination therapy did not affect the pharmacokinetic properties of either drug [34]. Brequinar is a potent inhibitor of DHODH and has been preclinically evaluated with anticancer effects in several solid tumor types [35,36]. However, it receives a low clinical efficacy and fails to gain FDA approval for cancer treatment. The lack of clinical efficacy may be caused by suboptimal dosing regimens of brequinar, which results in the failure to inhibit DHODH sustainably [37]. Thus, a combination therapy is essential for brequinar to augment its anticancer capacities. Despite several clinical trials of brequinar that have been performed on several cancers, not one has been carried out based on the combination with cisplatin. Our study proposes the possibility to use brequinar combined with cisplatin for treating recurrent and metastatic cervical cancer, which needs further trials to verify. Meanwhile, the underlying mechanism of a synergistic effect of combined therapy was also discussed. It is noted that the regulatory function of DHODH inhibition on the mTOR signaling remains controversial. A most recent study reveals that the activity of the mTOR pathway is strongly attenuated in *DHODH*-knockout medulloblastoma SU_MB002 cell lines, thereby inducing cell-cycle arrest and apoptosis [38]. Conversely, Hoxhaj et al. [39] fail to block the mTOR signaling in HeLa cells by using a DHODH inhibitor leflunomide, even after 24 h treatment. Surprisingly, the combination of cisplatin and DHODH inhibition significantly downregulated the mTOR pathway activity in our study, which is critically involved in inducing ferroptosis in cervical cancer cells [40,41]. In conclusion, the ferroptotic synergy may be increased through DHODH inhibition and mTOR pathway inhibition.

## 5. Conclusions

In conclusion, DHODH was upregulated in cervical cancer tissues. Silence of *DHODH* or pharmacologic inhibition promoted the ferroptotic death in cervical cancer cells. Moreover, inhibiting DHODH in cervical cancer cells both in vitro and in vivo produced a synergistic anticancer effect with cisplatin, which may be achieved by mTOR pathway suppression. Our work proposes that the combination of DHODH inhibition and cisplatin is a potential strategy for cervical cancer treatment.

## Figures and Tables

**Figure 1 cancers-15-00546-f001:**
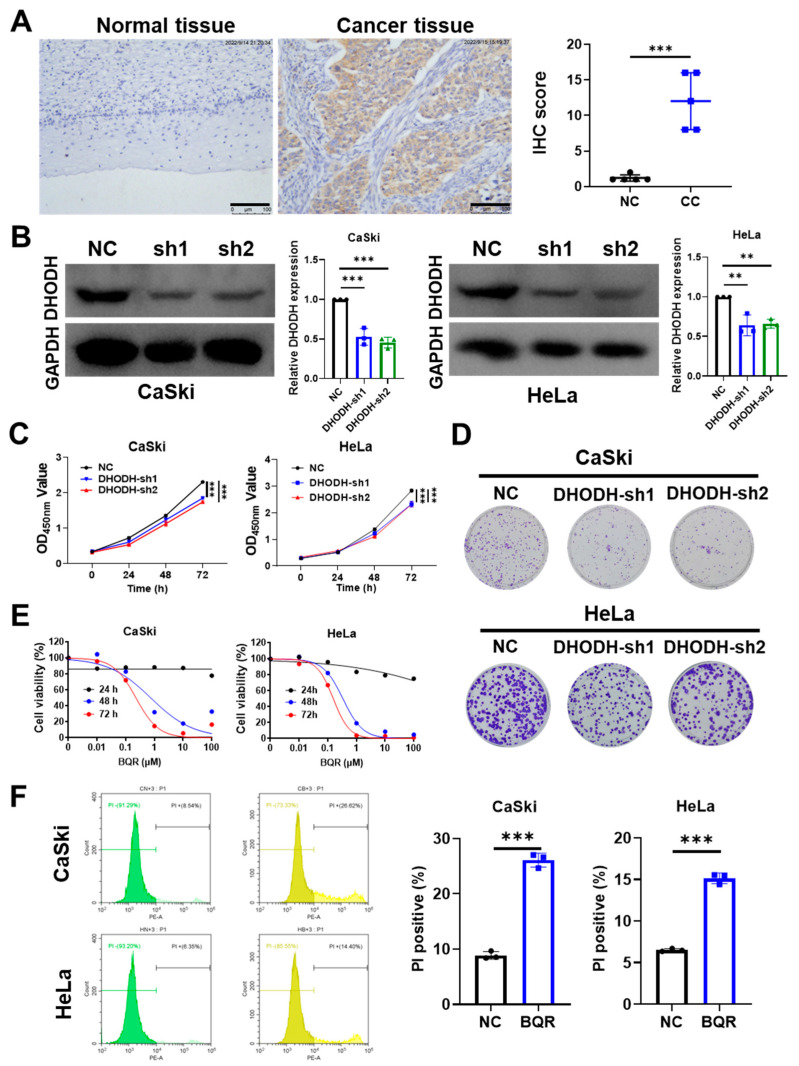
Inhibition of DHODH suppresses cell proliferation and promotes cell death in cervical cancer cells. (**A**) The expression level of DHODH protein in human cervical cancer tissues and adjacent non-tumor tissues was measured by IHC staining. Scale bar = 100 μm. IHC scores were used to analyze the experimental results. *** *p* < 0.001 compared to the normal tissues. (**B**) The efficacy of *DHODH* knockdown in CaSki and HeLa cells was measured by Western blotting. The relative expression of DHODH was calculated by ImageJ software in each group (*n* = 3). ** *p* < 0.01, *** *p* < 0.001 compared to the control group. (**C**) The effect of *DHODH* knockdown on cell viability was measured by CCK-8 assays in cervical cancer cells. *** *p* < 0.001 compared to the control group. (**D**) The effect of *DHODH* knockdown on clonogenicity was measured by the colony formation assay in cervical cancer cells. (**E**) Sensitivity to brequinar (BQR) was assessed using the CCK-8 assay in CaSki and HeLa cells treated with BQR (0, 0.01, 0.1, 1, 10, 100 μM) for 24, 48 and 72 h. (**F**) The effect of BQR (0.7 μM for CaSki cells; 0.3 μM for HeLa cells) on cell death was assessed by the flow cytometry analysis through PI staining. *** *p* < 0.001 compared to the control group.

**Figure 2 cancers-15-00546-f002:**
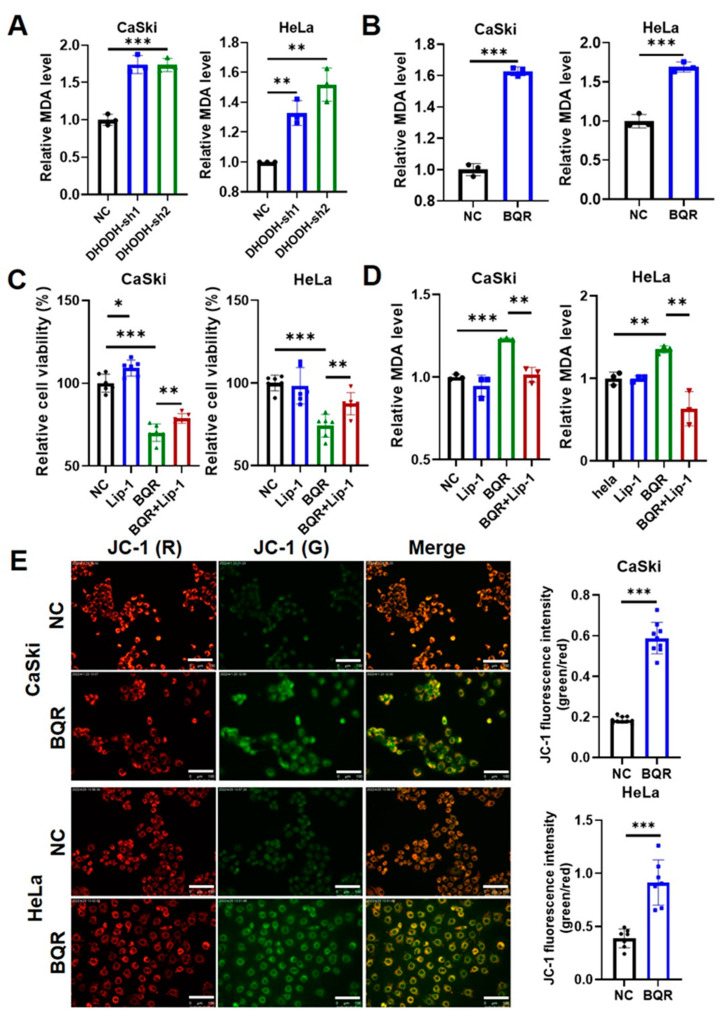
Inhibition of DHODH induces ferroptotic cell death in cervical cancer cells. (**A**) The effect of DHODH knockdown on lipid peroxidation in CaSki and HeLa cells was measured by MDA level detection. ** *p* < 0.01, *** *p* < 0.001 compared to the control group. (**B**) The effect of brequinar (BQR) on lipid peroxidation in CaSki (2 μM) and HeLa (1 μM) cells was measured by MDA level detection after treated for 48 h. *** *p* < 0.001 compared to the control group. (**C**) Cell viability of CaSki and HeLa cells treated with BQR (0.7 μM for CaSki cells; 0.3 μM for HeLa cells) and/or liproxstatin-1 (Lip-1) (2 μM) for 24 h was measured by CCK-8 assays. * *p* < 0.05, ** *p* < 0.01, *** *p* < 0.001. (**D**) The lipid peroxidation of CaSki and HeLa cells treated with BQR (0.7 μM for CaSki cells; 0.3 μM for HeLa cells) and/or Lip-1 (2 μM) for 24 h was measured by MDA level detection. ** *p* < 0.01, *** *p* < 0.001. (**E**) The mitochondrial dysfunction in CaSki and HeLa cells treated with or without BQR (0.7 μM for CaSki cells; 0.3 μM for HeLa cells) was measured by mitochondrial membrane potential detection through JC-1 staining. Scale bar = 100 μm. *** *p* < 0.001 compared to the control group.

**Figure 3 cancers-15-00546-f003:**
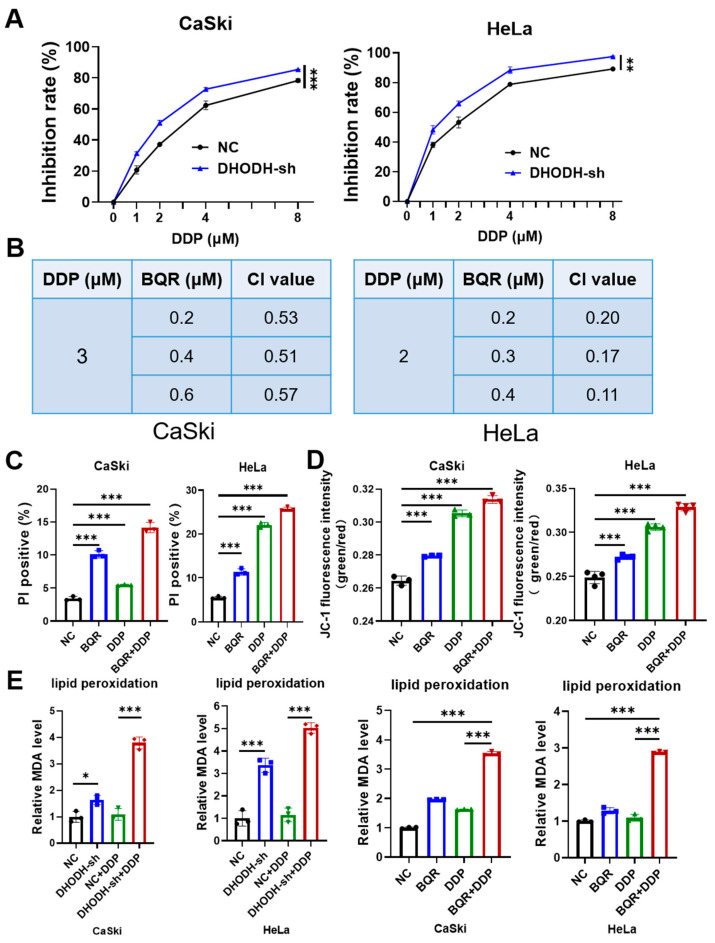
Brequinar (BQR) synergizes with cisplatin (DDP) to exert anticancer effects by inducing ferroptotic death in cervical cancer cells in vitro. (**A**) Cell viability of CaSki and HeLa cells treated with BQR (0.2 μM for CaSki cells; 0.1 μM for HeLa cells) and/or DDP (2 μM for CaSki cells; 1 μM for HeLa cells) for 48 h was measured by CCK-8 assays. ** *p* < 0.01, *** *p* < 0.001 compared to the control group. (**B**) The synergistic effect of BQR (0.2 μM for CaSki cells; 0.1 μM for HeLa cells) and DDP (2 μM for CaSki cells; 1 μM for HeLa cells) was measured in CaSki and HeLa cells. (**C**) The effect of combination of BQR (0.2 μM for CaSki cells; 0.1 μM for HeLa cells) and DDP (2 μM for CaSki cells; 1 μM for HeLa cells) on cell death in cervical cancer cells was assessed by flow cytometry analysis through PI staining. *** *p* < 0.001. (**D**) The mitochondrial dysfunction in CaSki and HeLa cells treated with BQR (0.2 μM for CaSki cells; 0.1 μM for HeLa cells) and/or DDP (2 μM for CaSki cells; 1 μM for HeLa cells) was measured by mitochondrial membrane potential detection through JC-1 staining. *** *p* < 0.001. (**E**) The synergistic effect of DHODH inhibition (*DHODH* knockdown or BQR (0.2 μM for CaSki cells; 0.1 μM for HeLa cells)) and DDP (2 μM for CaSki cells; 1 μM for HeLa cells) on lipid peroxidation was measured by MDA level detection in cervical cancer cells. * *p* < 0.05, *** *p* < 0.001.

**Figure 4 cancers-15-00546-f004:**
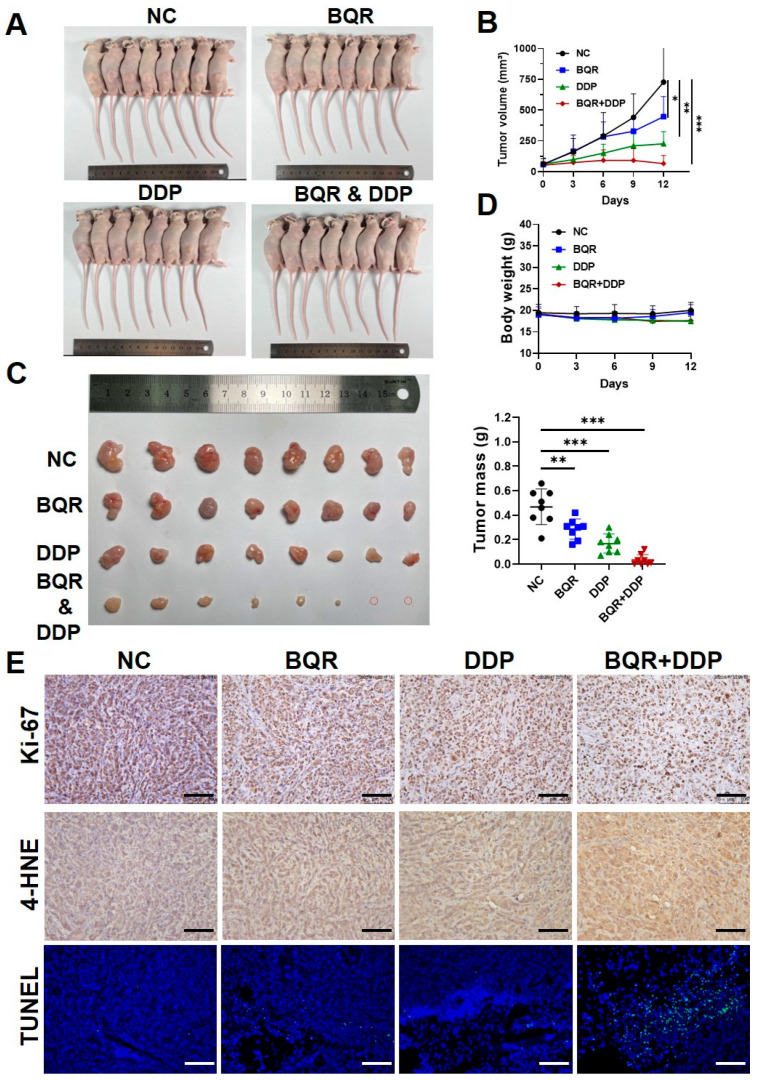
Brequinar (BQR) synergizes with cisplatin (DDP) to inhibit the growth of cervical cancer cells in vivo. (**A**) Images of nude mice with subcutaneous tumor injected by HeLa cells. Groups of mice were treated with BQR (15 mg/kg) and/or DDP (7 mg/kg) as indicated (*n* = 8 per group). (**B**) Growth curves of HeLa tumors of each group. * *p* < 0.05, ** *p* < 0.01, *** *p* < 0.001 compared to the control group. (**C**) Images of resected tumors from each group, and tumor mass of HeLa tumors of each group. ** *p* < 0.01, *** *p* < 0.001 compared to the control group. (**D**) Body weight of nude mice in each group. (**E**) Representative images of IHC staining for Ki-67 and 4-HNE, and representative images of TUNEL staining from sections of xenografted tumors. Scale bar = 250 μm.

**Figure 5 cancers-15-00546-f005:**
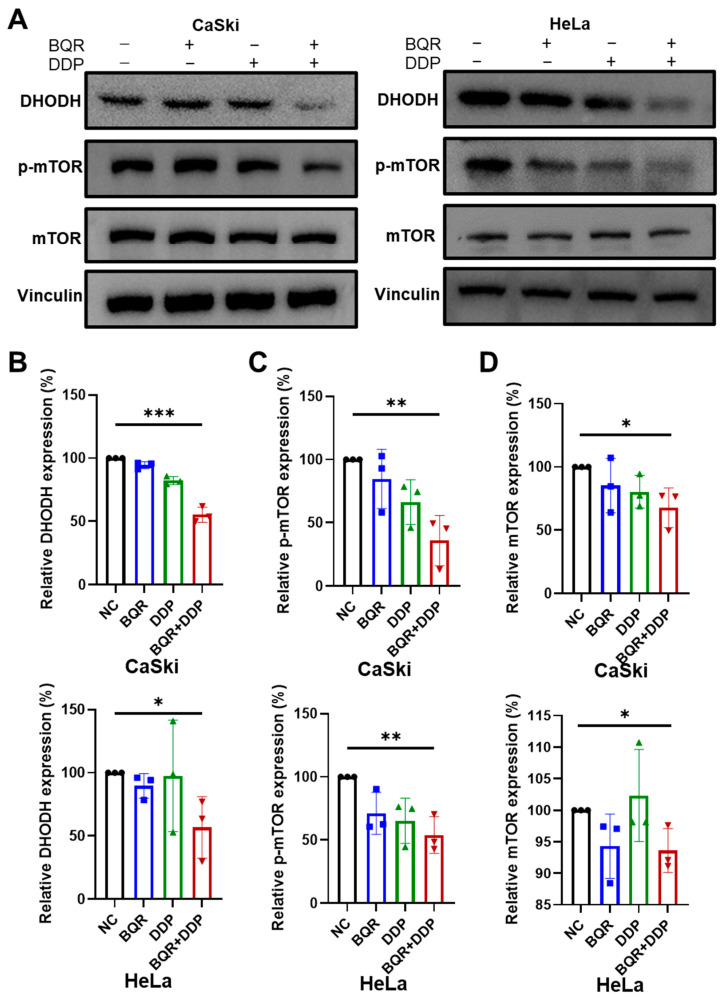
Brequinar (BQR) synergizes with cisplatin (DDP) to suppress the expression of DHODH via inhibiting mTOR pathway. (**A**) The levels of DHODH, p-mTOR and mTOR proteins were detected by Western blotting in CaSki and HeLa cells treated with BQR (0.2 μM for CaSki cells; 0.1 μM for HeLa cells) and/or DDP (2 μM for CaSki cells; 1 μM for HeLa cells) for 48 h. (**B**–**D**) Relative expression of DHODH, p-mTOR, and mTOR was calculated by ImageJ software in each group (*n* = 3). * *p* < 0.05, ** *p* < 0.01, *** *p* < 0.001.

## Data Availability

All the data supporting this study are available from the corresponding author upon reasonable request.

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
