# Peer review of "DHODH Inhibition Exerts Synergistic Therapeutic Effect with Cisplatin to Induce Ferroptosis in Cervical Cancer through Regulating mTOR Pathway"

_cancers, 2023, doi:10.3390/cancers15020546_

Round 1

Reviewer 1 Report

Dear authors,

Congratulations on your very interesting paper. The manuscript will make an impact in the field.

1. The manuscript could benefit from a more conventional writing for a larger audience.

2. In the abstract, please remove the headlines of the sections. 

Reviewer 2 Report

This is a very meaningful research. DHODH inhibition and DDP have synergistic anti-tumor effects, and this significance is closely related to iron death and mTOR pathway Question 1. The use of HPV vaccine and the screening of intraepithelial neoplasia have been widely carried out in economically developed regions of China. In economically underdeveloped regions of the world, cervical cancer is an important cause of serious threats to women‘s health. Therefore, China in LINE39 42 should be changed to developing countries and regions Question 2. Line 350 (in inclusion,.....): The inference of mTOR mechanism should be elaborated in the discussion.
